# The Evaluation of Vascular Endothelial Growth Factor A (VEGFA) and VEGFR2 Receptor as Prognostic Biomarkers in Bladder Cancer

**DOI:** 10.3390/diagnostics13081471

**Published:** 2023-04-19

**Authors:** Meryem El Azzouzi, Hajar El Ahanidi, Chaimae Hafidi Alaoui, Imane Chaoui, Laila Benbacer, Mohammed Tetou, Ilias Hassan, Mounia Bensaid, Mohamed Oukabli, Ahmed Ameur, Abderrahmane Al Bouzidi, Mohammed Attaleb, Mohammed El Mzibri

**Affiliations:** 1Biology and Medical Research Unit, Centre National de l’Energie, des Sciences et des Techniques Nucléaires (CNESTEN), Rabat 10001, Morocco; 2Faculty of Medicine and Pharmacy of Rabat, Mohammed V University in Rabat, Rabat 10100, Morocco; 3Department of Pathology and Immunology, Faculty of Medicine, University of Geneva, 1211 Geneva, Switzerland; 4Faculty of Sciences, Mohammed V University in Rabat, Rabat 10040, Morocco; 5Department of Urology, Mohammed V Military Teaching Hospital of Rabat, Rabat 10045, Morocco; 6Department of Pathology, Mohammed V Military Teaching Hospital of Rabat, Rabat 10045, Morocco

**Keywords:** bladder cancer, VEGF, VEGFR1/R2, biomarker, gene expression

## Abstract

Vascular endothelial growth factor (VEGF) and its receptors (VEGFR1 and VEGFR2) are the most important tissue factors involved in tumor growth and angiogenesis. The aim of this study was to evaluate the promoter mutational status of VEGFA and the expression levels of VEGFA, VEGFR1, and VEGFR2 in bladder cancer (BC) tissues and to correlate the results with the clinical–pathological parameters of BC patients. A total of 70 BC patients were recruited at the Urology Department of the Mohammed V Military Training Hospital in Rabat, Morocco. Sanger sequencing was performed to investigate the mutational status of VEGFA, and RT-QPCR was used to evaluate the expression levels of VEGFA, VEGFR1, and VEGFR2. Sequencing of the VEGFA gene promoter revealed the presence of −460T/C, −2578C/A, and −2549I/D polymorphisms, and statistical analyses showed a significant correlation between −460T/C SNP and smoking (*p* = 0.02). VEGFA and VEGFR2 expressions were significantly up-regulated in patients with NMIBC (*p* = 0.003) and MIBC (*p* = 0.03), respectively. Kaplan–Meier analyses showed that patients with high VEGFA expression had significantly longer disease-free survival (*p* = 0.014) and overall survival (*p* = 0.009). This study was very informative, showing the implication of VEGF alterations in BC, suggesting that VEGFA and VEGFR2 expressions could be promising biomarkers for the better management of BC.

## 1. Introduction

Worldwide, bladder cancer (BC) incidences continue to increase, due to the modifications in our behavior and changes in environmental, nutritional, and professional habits. Currently, BC is the 10th most common cancer, with 570,000 new cases per year and 210,000 deaths annually [1]. BC histological classification is based on the invasion of the muscularis in the bladder wall. Accordingly, 70–75% of cases are non-muscle invasive bladder cancer (NMIBC; includes Ta, T1 and Tis), and 25–30% are muscle invasive bladder cancer (MIBC; includes T2, T3 and T4) [2]. The standard care for MIBC patients consists of neoadjuvant chemotherapy followed by radical cystectomy. Patients with NMIBC can generally be managed with transurethral resection of the bladder (TURB) associated with intravesical chemotherapy (mitomycin C) or immunotherapy (Bacillus Calmette–Guerin therapy). However, within two years of initial diagnosis, 70–85% of NMIBC cases will recur, and 15–30% will progress to MIBC [3]. Therefore, more efforts are required to determine a molecular signature, with high diagnosis and prognosis values, to improve the monitoring and the management of tumor recurrence and prognosis prediction in BC [4].

Angiogenesis is an important mechanism in the development, progression, and metastasis of different cancers, and it is essential to supply the oxygen and nutrients for tumor growth. In solid tumors, including BC, hypoxia was reported to correlate with poor prognoses and low survival rates in cancer patients [5]. The vascular endothelial growth factor (VEGF) is the major regulator of angiogenesis, which contributes to key aspects in the tumor microenvironment, including the function of cancer stem cells and tumor initiation [6]. VEGFA is identified by the ability to stimulate the vascular endothelial cell proliferation and to increase the vascular permeability. It also promotes the migration and survival of endothelial cells [7]. Recently, there are growing number of studies addressing the association between VEGFA gene SNPs and various cancer types, including urogenital carcinomas [8] and lung cancer [9]. Several polymorphisms in the *VEGFA* gene, especially in its promoter region, such as −460C/T (rs833061); −2578C/A (rs699947); and −2549I/D (rs35569394) have been widely reported to contribute to tumor growth and progression and may alter the expression level of VEGFA in various malignancies [10].

VEGF ligands bind to VEGF receptors in order to initiate the pathway of angiogenesis and vascular permeability [11]. The VEGF/VEGFR axis has different biological effects on cancer cells. In neo-vascularization, VEGFR1 and VEGFR2 act as receptors for VEGFA [7], and an increased level of VEGFA expression is found in BC specimens. However, limited studies have been conducted to evaluate the expression of VEGF in combination with the expression statuses of VEGFR1 and VEGFR2 receptors in bladder tumors and to assess its clinical importance on disease recurrence and prognosis.

Thus, this study was planned to determine the mutational status of VEGFA and the expression levels of *VEGFA*, *VEGFR1*, and *VEGFR2* genes to identify potential recurrence and prognostic prediction biomarkers for improving BC management.

## 2. Material and Methods

### 2.1. Population Study

A number of 70 BC patients were recruited from the Military Hospital Mohamed V in Rabat, Morocco (Urology Department). Staging and grading were determined according to the TNM classification and World Health Organization criteria [12]. All patients were followed up in accordance with standard recommendations [3]. Recurrence was the reappearance of primary NMIBC with a lower or the same stage, and progression was a disease with a higher TNM stage upon relapse. Disease-free survival (DFS) was the time from surgical resection (TURB) to local recurrence, death, or last follow-up, and overall survival (OS) was the time from diagnosis to death from any cause or last follow-up.

The protocol of the study was approved by the Ethics Committee for Biomedical Research, Faculty of Medicine and Pharmacy of Rabat, Morocco (Ref 82/19), and written informed consent was obtained from all patients.

### 2.2. DNA and RNA Extraction

Genomic DNA was extracted from fresh biopsies using the phenol/chloroform approach [13]. Total RNA was extracted from fresh-frozen biopsies, which were conserved in RNAlater (Invitrogen), using TRIsol Reagent (Sigma-Aldrich, Inc., St. Louis, MO, USA) according to the manufacturer’s instructions. All extracted DNAs and RNAs were quantified by NanoDrop 2000 (Thermo Fisher Scientific, Waltham, MA, USA).

### 2.3. Evaluation of VEGFA Mutational Status

The mutational status was assessed by PCR-sequencing. The VEGFA mutations, namely rs833061, rs699947, and rs35569394, were identified on two separate PCR products amplified using specific primers (Table 1). The amplification reaction was carried out in a final volume of 25 µL containing 1× PCR buffer, 10 mM of each dNTP, 1.5 mM MgCl_2_, 10 µM of each primer, 1U of Platinum Taq Polymerase (Invitrogen), and 2 µL of DNA. Mixtures were first denatured at 95 °C for 7 min. Then, 35 cycles of PCR were performed with denaturation at 95 °C for 30 sec, primer annealing for 1 min at 60 °C, and primer extension for 30 s at 72 °C. At the end of the last cycle, the mixtures were incubated at 72 °C for 7 min. For every reaction a negative control was included. Purification of PCR products was performed using the Ex’S-Pure™ enzymatic PCR cleanup kit (Nimagen). Sequencing reaction was performed in a final volume of 10 µL containing 10 µM of forward or reverse primer (the same used for PCR amplification), 1 µL of 2.5× Big Dye ready reaction mix v.3.1, and 100 ng of purified PCR product. The mixtures were incubated at 96 °C for 1 min, and 25 cycles were performed: denaturation at 96 °C for 10 s, primer annealing at 50 °C for 5 s, and extension at 60 °C for 4 min. Sequencing reaction products were finally purified using Sephadex G-50 gel-exclusion chromatography (GE Healthcare Life Science., St. Louis, MO, USA). DNAs were then sequenced on an ABI 3130XL DNA analyzer, and the obtained sequences were matched with the gene reference sequences collected from the GenBank database (NCBI Reference Sequence: NG_008732.1). The sequence alignments were performed using Sequencing Analysis V5.4 Software.

### 2.4. Gene Expression Study

Total RNA was subjected to reverse transcriptase reaction using the high-capacity cDNA reverse transcription kit (Applied Biosystems, Cheshire, UK). cDNAs were subsequently used to perform a SYBR Green-based qRT-PCR by the PowerUp SYBR Green Master Mix (Thermo Fisher Scientific). Samples were amplified in triplicate in one-assay runs, with a non-template control for each primer pair to control for primer dimerization or for contamination. VEGFA, VEGFR1, and VEGFR2 levels were normalized to the expression of β2 microglobulin (β2M), used as an internal control gene, using the 2^−ΔCt^ formula. Primer sequences used for VEGFA, VEGFR1, VEGFR2, and β2M quantification are reported in Table 1.

### 2.5. Statistical Analysis

The statistical analyses were performed using Graph Pad Prism software version 9 and SPSS software version 23. The correlation between the mutational status of the *VEGFA* promoter and the clinico-histopathological features (age of patients, smoking status, and tumor’s stage, grade, and recurrence/progression) were evaluated by the χ^2^ test (*p* values < 0.05 were considered as statistically significant). For the comparison of multiple groups, non-parametric tests (Mann–Whitney or Kruskal–Wallis tests) were used. Survival analyses were realized by the Kaplan–Meier method and compared using a log-rank test; *p*-values < 0.05 were interpreted as significant and labeled with a *. Univariate and multivariate analyses were performed using the Cox proportional hazards regression models to identify the prognostic factors influencing DFS and OS. Results were expressed as hazard ratios (HRs) with their 95% confidence intervals (CIs). The prognostic factors with *p*-values < 0.2 in the univariate model were further entered into the multivariate analysis. The Cox proportional hazard model was further used for multivariate analysis and included the following variables: age (<50, 50–70 and >70 years), tumor staging (NMIBC vs. MIBC), tumor grading (low grade vs high grade), and VEGFA expression (low expression vs high expression). The results were presented as hazard ratios (HRs) with 95% confidence intervals (CIs). Statistical significance was assumed at *p*-value < 0.05.

## 3. Results

### 3.1. Characteristics of the Study Population

Among the 70 recruited patients, 68 were male, and only 2 were female. The mean age of patients was 67 years (from 47 to 85 years). Clinico-pathological characteristics of recruited patients are summarized in Table 2; most cases had NMIBC stages (≤PT1) (74.3%) and high tumor grades (61.4%). In the present study, tumor recurrence and progression were studied only in NMIBC cases. Accordingly, among the 52 NMIBC cases, 23.1% recurred (12/52), and 9.6% progressed (5/52) (Table 2).

### 3.2. Distribution of Genotype and Allelic Frequencies of VEGFA Polymorphisms

Sequencing analysis revealed the presence, in our cohort, of the three targeted polymorphisms: rs833061 (−460 T/C); rs699947 (−2578 C/A); and rs35569394 (−2549 I/D). The genotype and allele frequencies of *VEGFA* polymorphisms in BC patients are summarized in Table 3. For the rs833061 SNP, the most prevalent genotype in BC cases was CC, reported in 51.4% of cases, and consequently, the C allele was the most prevalent allele identified in 62.1% of cases. For the rs699947 SNP, the AC genotype prevailed and was reported in 41.1% of cases; A and C alleles were identified in 52.1% and 47.9% of cases, respectively. For the rs35569394 SNP, approximate genotype frequencies were reported in BC cases; the allele containing the insertion was identified in 50.7% of cases, whereas 49.3% showed the allele harboring the deletion.

### 3.3. Association of VEGFA Polymorphisms and Clinico-Pathological Features of BC Patients

The associations between genotype and allele frequencies of VEGFA polymorphisms (−460 T/C; −2578 C/A; and −2549 I/D) and clinico-pathological parameters of BC patients were investigated, and the results are reported in the Appendix A (Appendix A, respectively). VEGFA genotypes were unrelated to the clinico-pathological features, such as gender, age, stage, grade, recurrence, and progression (*p* > 0.05). However, smoking was significantly associated with the −460 T/C polymorphism (*p* = 0.02) (Appendix A). Moreover, a borderline significance was also found between smoking and the −2578 C/A polymorphism (*p* = 0.07) (Appendix A).

Of particular interest, a very significant association was detected between the −2578 C/A and −2549 I/D polymorphisms. The CC homozygous genotype of the rs699947 polymorphism was mostly detected in patients harboring the deletion genotype of the rs35569394 polymorphism. Likewise, the heterozygous CA and the homozygous AA genotypes of the rs699947 polymorphism were predominantly detected, respectively, in patients with I/D and II genotypes at the rs35569394 site (*p* = 0.001).

### 3.4. The Relationship between the VEGFA Expression and Clinico-pathological Parameters of BC Patients

The expression of VGEFA was evaluated in all BC cases, and the associations between VEGFA expression and clinico-pathological features (tumor stage, histological grade and follow-up) are reported in Figure 1. Analyses of qRT-PCR data showed that VEGFA expression was significantly higher in NMIBC tumors (≤PT1), compared to MIBC (>PT1) (*p* = 0.003) (Figure 1A). VEGFA expression was up-regulated in high grade (Figure 1B) and was slightly similar between primary and recurrence tumors (Figure 1C), but statistical analyses showed no significant differences; *p*-values ≥ 0.05.

### 3.5. Association between the VEGFA Genotypes and VEGFA mRNA Expression in BC

In the present study, the impact of *VEGFA* gene polymorphisms on VEGFA expression was also assessed, and the results are presented in Figure 2. The expression level of VEGFA was shown to be slightly higher in patients with the CT genotype of rs833061 SNP, in AA and AC genotypes of rs699947 SNP, and in II and ID of rs35569394 SNP. However, statistical analysis showed that VEGFA expression was not significantly associated with *VEGFA* gene polymorphisms (*p* > 0.05).

### 3.6. The Relationship between VEGF Receptor (VEGFR1/R2) Expressions and Clinical Pathological Features of BC Patients

VEGFR1 and VEGFR2 mRNA expression levels were also investigated according to the clinico-pathological features of BC cases, and the results are reported in Figure 3. Overall, VEGFR1 was more expressed in NMIBC cases, and VEGFR2 was more expressed in MIBC cases. The expression of VEGFR1 was unrelated to tumor grades, stages, and the follow-up status (*p* > 0.05) (Figure 3). The expression of VEGFR2 was significantly associated with tumor stages (*p* = 0.03 < 0.05) (Figure 3A). No significant difference was reported between VEGFR2 expression and tumor grade and patient follow-up (*p* ≥ 0.05) (Figure 3B,C).

### 3.7. The Evaluation of VEGFA, VEGR1, and VEGFR2 Expressions as a Prognosis Survival Biomarker in BC

To evaluate whether the expression of studied genes has an impact on patients’ clinical outcomes (disease-free survival (DFS) and overall survival (OS)), Kaplan–Meier analyses were performed, and the results are reported in Figure 4. The mean and median of clinical follow-up periods for this cohort were 44 and 45 months, respectively. Based on the median value of the gene expression results, patients were divided into two groups: low- and high-expression groups. Interestingly, patients with high VEGFA expression had significantly longer DFS (*p* = 0.014) and OS (*p* = 0.009) than those with low expressions (Figure 4A,B) (Table 4). For VEGFR1 and VEGFR2, the DFS and OS were almost the same (Figure 4C–F). The survival curves showed no statically significant differences between the ‘low-expression’ and the ‘high-expression’ groups of VEGFR1 (DFS: *p* = 0.522; OS: *p* = 0.412) and VEGFR2 (DFS: *p* = 0.698; OS: *p* = 0.859, respectively).

## 4. Discussion

Angiogenesis represents a critical process in the growth and progression of various solid malignancies. Therefore, the molecular mechanisms involved in angiogenesis show a particular interest in cancer research to identify promising biomarkers for better cancer management. In this field, growing interest is given to the study of *VEGFA* gene polymorphisms in many kinds of tumors [8,14].

In this study, we aimed to determine the mutational status of VEGFA and the expression levels of VEGFA, VEGFR1, and VEGFR2 genes in order to identify potential biomarkers for improving BC management. In our cohort, the majority of patients were male, which might be due to the high incidence of smoking among men and the burden of environmental, occupational, and professional exposures, which are considered the main causes of BC [15].

In the present study, our interest was focused on three SNPs in the promoter region of the VEGFA gene, −460 T/C, −2578 C/A, and −2549 I/D, widely reported to be associated with many cancer developments and progressions [10,16]. In our findings, a significant association between −460 T/C SNP and smoking was reported. These results are in agreement with other studies reporting an important interaction between the −460 T/C polymorphism and smoking in esophageal adenocarcinomas [17] and prostate cancer [18].

Aside from that, no statistically significant correlation was revealed between VEGFA polymorphisms and BC parameters; these results confirm previously reported data highlighting the absence of correlations between −460 T/C and −2549 I/D polymorphisms and BC staging, grading, and tumor progression [6,19]. However, a recent meta-analysis showed significant correlations between the −2578 C/A polymorphism and histological grade and tumor stage in BC [16].

In this study, no significant association was found between these SNPs and clinical outcomes evaluated by the recurrence and progression rates. These results corroborate with a previous study that showed the absence of correlations between VEGFA polymorphisms and tumor recurrence and progression [20].

Of particular interest, a strong association was revealed between the −2578 A/C and −2549 I/D SNPs. This linkage disequilibrium between the two loci was already reported on Alzheimer’s and BC cases, which found that individuals with the rs699947-A allele had an 18 bp insertion (rs35569394-I), while those with the rs699947-C allele had a deletion (rs35569394-D) [21,22]. This strong association is due to the small distance between the two loci, limiting the independent segregation of the corresponding alleles.

In this study, the association between the expression level of VEGFA and the clinico-pathological features of BC was also investigated. Higher expression levels were detected in NMIBC stages, high grades, and recurrent cases, but a significant association was detected only with tumor stage. Increased VEGFA expression levels were reported even in superficial tumors, compared to the normal urothelium [5]. In a previous study, Inoue et al. (2000) reported a strong association between VEGFA expression and advanced grade, stage, and recurrence of urothelial cell carcinoma [23]. Poyet et al. (2017) showed the involvement of VEGFA expression in the progression of NMIBC tumors into MIBC [24]. This difference between our results, mainly regarding the distribution of VEGFA expression according to the tumor stage, might be due to the small sample size, as well as the large heterogeneity of VEGFA expression between BC cases.

Currently, it is widely accepted that several polymorphisms in the VEGFA promoter are reported to affect the gene expression [6]. Thus, we evaluated the genetic association between VEGFA SNPs (−460 T/C, −2578 C/A, and −2549 I/D) and the gene expression profile. Overall, no significant correlation was revealed between these SNPs and the gene expression level. In contrast, Jaiswal et al. (2013) found that −2578 C/A genotypes are associated with higher expression levels of VEGFA [19]. Moreover, Chen et al. (2020) related the two polymorphisms, −460 T/C and −2578 C/A, to higher VEGF expression in BC [6]. Similar to VEGF, increased VEGFR expression was also found in BC [10]. In our study, the evaluation of the correlation between VEGFR1/VEGFR2 expression levels and BC parameters showed a significantly high level of VEGFR2 expression in MIBC. This result is in agreement with the study of Kopparapu et al. (2013), which found, by immunohistochemistry staining, that VEGFR2 expression was significantly higher in advanced bladder tumors [11].

In the current study, survival analysis showed a better DFS and OS in patients displaying higher VEGFA expression than in patients with lower expression. However, no impacts of VEGFR1 and VEGFR2 gene expressions on DFS and OS were revealed. Another study showed that patients with higher expressions of VEGFA and VEGFR1 tended to have a poorer DFS, but the results were not statistically significant, compared with those with lower expressions [11].

Overall, this study showed the involvement of the VEGFA SNPs −460 T/C, −2578 C/A, and −2549 I/D in BC and suggested that VEGFA and VEGFR2 gene expressions might be promising prognostic biomarkers for the better management of BC. However, the relatively small number of patients and the small number of progressive/recurrent cases present limitations for this study. Further studies in larger cohorts are needed to confirm our results.

## 5. Conclusions

The expression of VEGFA is not affected by the −460 T/C, −2578 C/A, and −2549 I/D SNPs, and high expression of VEGFA and its receptor, VEGFR2, are significantly associated with BC staging, suggesting that they can be used as promising biomarkers for BC diagnosis. Moreover, the strong association between VEGFA expression and both DFS and OS is in favor of using VEGFA as a prognostic biomarker. Thus, VEGFA and VEGFR2 expressions are likely promising biomarkers for the better management of BC.

## Figures and Tables

**Figure 1 diagnostics-13-01471-f001:**
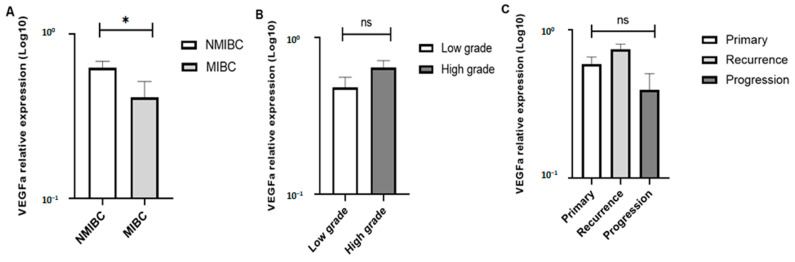
VEGFA expression levels according to tumor staging (**A**), grading (**B**), and follow-up (**C**). *: *p* values < 0.05; ns (not significant): *p* values ≥ 0.05. (**B**): Tumor grading was performed for NMIBC and MIBC cases; (**C**): Progression and/or recurrence rates were performed only for NMIBC cases.

**Figure 2 diagnostics-13-01471-f002:**
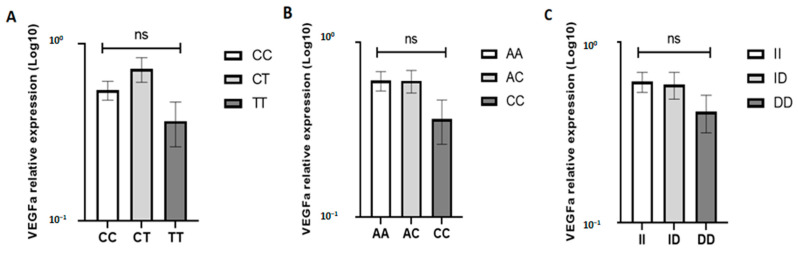
Association between the VEGFA mRNA expression level and the VEGFA SNP genotypes of −460 C/T (**A**), −2578 C/A (**B**), and −2549 I/D (**C**). *p*-values ≥ 0.05 were labeled with ns (not significant).

**Figure 3 diagnostics-13-01471-f003:**
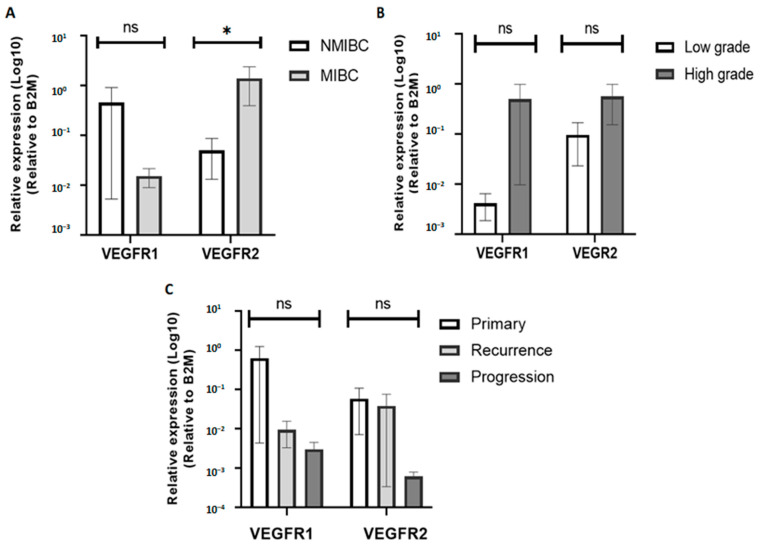
VEGFR1 and VEGFR2 mRNA expressions according to tumor staging (**A**), grading (**B**), and follow-up (**C**). *: *p* values < 0.05; ns (not significant): *p* values ≥ 0.05.

**Figure 4 diagnostics-13-01471-f004:**
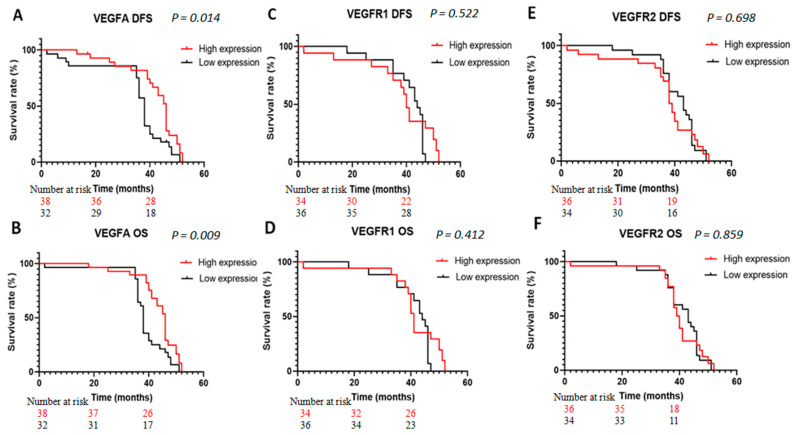
Kaplan–Meier analyses of disease-free survival (DFS) and overall survival (OS) correlated to VEGFA (**A**,**B**), VEGFR1 (**C**,**D**), and VEGFR2 (**E**,**F**) expressions in bladder cancer patients.

**Table 1 diagnostics-13-01471-t001:** Primers’ sequences used for VEGFA mutational status assessment and VEGFA, VEGFR1, and VEGFR2 expression analyses.

Analysis	Forward Primer 5′–3′	Reverse Primer 5′–3′
VEGFA mutational status		
rs833061 rs699947 and rs35569394	TGTGCAGACGGCAGTCACTA	CCCGCTACCAGCCGACTTT
	ATAAGGGCCTTAGGACACCA	GCTACTTCTCCAGGCTCACA
VEGFA and VEGFR1/R2 expression analyses		
B2M expression	GAGGCTATCCAGCGTACTCCA	CGGCAGGCATACTCATCTTTT
VEGFA expression	AGGGCAGAATCATCACGAAGT	AGGGTCTCGATTGGATGGCA
VEGFR1 expression	GAAAACGCATAATCTGGGACAGT	GCGTGGTGTGCTTATTTGGA
VEGFR2 expression	GTGATCGGAAATGACACTGGAG	CATGTTGGTCACTAACAGAAGCA

**Table 2 diagnostics-13-01471-t002:** Tumor characteristics of recruited cases.

Parameter	Total Cases	Percentage (%)
Gender		
Male	68	97.1
Female	2	2.9
Age		
<50	1	1.4
50–70	44	62.9
>70	25	35.7
Smoking		
Yes	28	40
No	42	60
Stage		
≤PT1	52	74.3
>PT1	18	25.7
Grade		
Low grade	27	38.6
High grade	43	61.4
Recurrence		
Yes	12	23.1
No	40	76.9
Progression		
Yes	5	9.6
No	47	90.4

**Table 3 diagnostics-13-01471-t003:** Genotypic and allelic frequency distributions of VEGFA polymorphisms in the study population.

Genotypes	−460 T/C		Genotypes	−2578 C/A		Genotypes	−2549 I/D	
	N	%		N	%		N	%
CC	36	51.4	AA	22	31.4	II	23	32.9
CT	15	27.2	AC	29	41.4	ID	25	35.7
TT	19	21.4	CC	19	27.2	DD	22	31.4
Alleles	−460 T/C		Alleles	−2578 C/A		Alleles	−2549 I/D	
	N	%		N	%		N	%
C	87	62.1	A	73	52.1	I	71	50.7
T	53	37.9	C	67	47.9	D	69	49.3

**Table 4 diagnostics-13-01471-t004:** Univariate and multivariate Cox regression results for the associations between VEGFA, VEGFR1, and VEGFR2 expressions and disease-free survival (DFS) and overall survival (OS) for BC.

Variable	Disease-Free Survival				Overall Survival			
	Univariate		Multivariate		Univariate		Multivariate	
	HR (95% CI)	*p*-Value	HR (95% CI)	*p*-Value	HR (95% CI)	*p*-Value	HR (95% CI)	*p*-Value
VEGFA expression	1.72 (1.03–2.88)	0.01	1.74 (1.038–2.93)	0.02	1.83 (1.1–3.1)	0.009	1.87 (1.12–3.14)	0.001
VEGFR1 expression	1.18 (0.70–1.99)	0.53	-	-	1.2 (0.71–2.03)	0.49	-	-
VEGFR2 expression	0.70 (0.39–1.27)	0.24	-	-	0.69 (0.38–1.24)	0.21	-	-

## Data Availability

There is no new data was created.

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
