# Peer review of "The Evaluation of Vascular Endothelial Growth Factor A (VEGFA) and VEGFR2 Receptor as Prognostic Biomarkers in Bladder Cancer"

_diagnostics, 2023, doi:10.3390/diagnostics13081471_

Round 1

Reviewer 1 Report

In this article, Meryem El Azzouzi, Hajar El Ahanidi et al. study the prognostic value of Vascular Endothelial Growth Factor A (VEGFA) and VEGFR2 receptor in bladder cancer.

General comment:

The topic of the study is relevant. However, in addition to the Kaplan-Meier estimates, the authors are encouraged to do uni- and multivariate Cox regression analysis to establish the prognostic value of VEGFA and its receptors in bladder cancer.

Minor comments:

Line 137, ‘Table 1. R2 expression analyses’: it should be indicated what ‘R2’ is.

Are the patients in Figure 1B and C those with NMIBC or MIBC or both? This should be indicated.

Figure 4: Number at risk could be added.

Author Response

Responses to Reviewer 1:

General comment:

The topic of the study is relevant. However, in addition to the Kaplan-Meier estimates, the authors are encouraged to do uni- and multivariate Cox regression analysis to establish the prognostic value of VEGFA and its receptors in bladder cancer.

As suggested, uni- and multivariate Cox regression analyses were done and added in the revised version of the manuscript. We really thank the reviewer for this relevant remark as these analyses will have more value to the study

Minor comments:

Line 137, Table 1 R2 expression analyses’: it should be indicated what ‘R2’ is.

As mentioned in the main document, VEGFA has two receptors VEGFR1 and VEGFR2, and Table 1 reports the sequences of primers used for VEGFA mutational status assessment and for VEGFA, VEGFR1 and VEGFR2 expression analyses. To avoid any misunderstanding the title of the Table was modified in the revised version as fellow: “Primers’ sequences used for VEGFA mutational status assessment and VEGFA, VEGFR1 and VEGFR2 expression analyses”.

Are the patients in Figure 1B and C those with NMIBC or MIBC or both? This should be indicated.

Thank you for this important remark. Tumor grading was done for both NMIBC and MIBC cases but recurrence and/or progression was assessed only for NMIBC cases. Accordingly, a footnote was added to the Figure 1 to give this information.

Figure 4: Number at risk could be added.

As requested, numbers at risk were added to the figure 4.

Reviewer 2 Report

The authors are presenting an interesting study on one of the most contemporary topics in uro-oncology - biomarkers in bladder cancer. the authors are presenting research on three SNPs in VEGFA and expression levels of VEGFA, VEGR1 and VEGR2 in BC biopsies, their interaction and their correlation with pathological and clinical variables in study population. Their results show promising results especially on VEGFA as prognostic marker, corelating with DFS and OS, as well as VEGFA and VEGFR2 expression showing statistically significant differences between NMIBC and MIBC.

Several issues need additional attention from the authors as follows:

Introduction: raw 45-46 - mitomycin is not standard of care for neoadjuvant chemotherapy - a typo?

Materials and methods - no changes needed

Results - raw 180-181 MIBC is not >=pT1 , it is > pT1

Discussion - a quick note must be made on the vast predominance of male patients - reason? possibly the type of hospital serving the military.

raw 285-289 should be re-written completely - it is not proper for the authors to assess their own study as 'very informative and has much strength', even more been followed by the authors own assessment of the two major limitations of the study - low number of patients and low percentage of progression and recurrence during the study timeframe.

The conclusions are adequately substantiated and represent a solid base to further investigation on the subject.  

 The references are appropriate and relevant to the subject and my recommendation is to accept this manuscript for publication after comments from the authors on the aforementioned issues.  

Author Response

Introduction: raw 45-46 - mitomycin is not standard of care for neoadjuvant chemotherapy - a typo?

We apologize for this mistake; it has been corrected in the revised version of the manuscript. In fact, the mitomycin is used as intravesical therapy for NMIBC cases. However, the MIBC are treated by the neoadjuvant chemotherapy followed by Cystectomy.

Results - raw 180-181 MIBC is not >=pT1, it is > pT1

We apologize for this typing mistake; it has been corrected in the revised version of the manuscript.

Discussion - a quick note must be made on the vast predominance of male patients - reason? possibly the type of hospital serving the military.

Thank you for this important remark. In Morocco, as it’s the case in many countries around the world, BC prevails among men and it’s due to the high incidence of smoking among men and the burden of environmental, occupational and professional exposures, considered as the main causes of BC. Moreover, Mohamed V military hospital hosts both military and civil patients and the prevalence of BC in men is not depending on depending on the military status of patients. As requested, the Discussion part of the revised version was revised to highlight why BC is more prevalent in men.

raw 285-289 should be re-written completely - it is not proper for the authors to assess their own study as 'very informative and has much strength', even more been followed by the authors own assessment of the two major limitations of the study - low number of patients and low percentage of progression and recurrence during the study timeframe

As requested, this paragraph of the discussion was re-edited and rephrased to be in line with the reviewer recommendation.